# Levels of Plasma Endothelin-1, Circulating Endothelial Cells, Endothelial Progenitor Cells, and Cytokines after Cardiopulmonary Bypass in Children with Congenital Heart Disease: Role of Endothelin-1 Regulation

**DOI:** 10.3390/ijms25168895

**Published:** 2024-08-15

**Authors:** Angélica Rangel-López, Héctor González-Cabello, María Eugenia Paniagua-Medina, Ricardo López-Romero, Lourdes Andrea Arriaga-Pizano, Miguel Lozano-Ramírez, Juan José Pérez-Barragán, Horacio Márquez-González, Dulce María López-Sánchez, Minerva Mata-Rocha, Ramon Paniagua-Sierra, Abraham Majluf-Cruz, Dina Villanueva-García, Sergio Zavala-Vega, Juan Carlos Núñez-Enríquez, Juan Manuel Mejía-Aranguré, José Arellano-Galindo

**Affiliations:** 1Unidad de Investigación Médica en Enfermedades Nefrológicas, UMAE Hospital de Especialidades, Centro Médico Nacional (CMN) Siglo XXI (SXXI), Instituto Mexicano del Seguro Social (IMSS), Mexico City 06720, Mexico; angelica.rangell@imss.gob.mx (A.R.-L.); miguelforenseadn@gmail.com (M.L.-R.); jose.paniagua@imss.gob.mx (R.P.-S.); 2Unidad de Investigación en Enfermedades Infecciosas, Hospital Infantil de México Federico Gómez-Secretaría de Salud (SS), Mexico City 06720, Mexico; 3Departamento de Neonatología e Infantes, UMAE Hospital de Pediatría, CMN SXXI IMSS, Mexico City 06720, Mexico; hector.gonzalezc@imss.gob.mx (H.G.-C.); ariesmd210379@hotmail.com (J.J.P.-B.); 4Departamento de Trasplantes, UMAE Hospital de Pediatría, CMN SXXI IMSS, Mexico City 06720, Mexico; paniaguamaria2@gmail.com; 5Unidad de Investigación en Biomedicina y Oncología Genómica, Hospital de Gineco-Pediatría 3A, IMSS, Mexico City 07760, Mexico; ricardolopez007@gmail.com; 6Unidad de Investigación en Inmunoquímica, UMAE Hospital de Especialidades, CMN SXXI IMSS, Mexico City 06720, Mexico; landapi@hotmail.com; 7Servicio de Cardiopatías Congénitas-UMAE Hospital de Cardiología, CMN SXXI IMSS, Mexico City 06720, Mexico; h.marquez@himfg.edu.mx; 8Departamento de Investigación Clínica, Hospital Infantil de México Federico Gómez, SS, Mexico City 06720, Mexico; 9Unidad de Investigación Médica en Epidemiología Clínica, UMAE Hospital de Pediatría, CMN SXXI IMSS, Mexico City 06720, Mexico; dulce.lopez@cieni.org.mx (D.M.L.-S.); juan.nuneze@imss.gob.mx (J.C.N.-E.); 10Centro de Investigación en Enfermedades Infecciosas, Instituto Nacional de Enfermedades Respiratorias, SS, Mexico City 14080, Mexico; 11Unidad de Investigación Médica en Genética Humana-UMAE Hospital de Pediatría, CMN SXXI IMSS, Mexico City 06720, Mexico; mmata@conahcyt.mx; 12Unidad de Investigación Médica en Hemostasia, Trombosis y Aterogénesis, Hospital General Regional 1, IMSS, Mexico City 03103, Mexico; amajlufc@imss.gob.mx; 13División de Neonatología, Hospital Infantil de México Federico Gómez, SS, Mexico City 06720, Mexico; dvillanueva@himfg.edu.mx; 14Laboratorio Clínico y Banco de Sangre, Instituto Nacional de Neurología y Neurocirugía, SS, Mexico City 14269, Mexico; sergio.zavala@innn.edu.mx; 15Laboratorio de Genómica del Cáncer, Instituto Nacional de Medicina Genómica, SS, Mexico City 14610, Mexico

**Keywords:** endothelin-1, circulating endothelial cells, endothelial progenitor cells, cytokines, congenital heart disease, cardiopulmonary bypass surgery, children, biomarkers

## Abstract

Congenital heart disease (CHD) can be complicated by pulmonary arterial hypertension (PAH). Cardiopulmonary bypass (CPB) for corrective surgery may cause endothelial dysfunction, involving endothelin-1 (ET-1), circulating endothelial cells (CECs), and endothelial progenitor cells (EPCs). These markers can gauge disease severity, but their levels in children’s peripheral blood still lack consensus for prognostic value. The aim of our study was to investigate changes in ET-1, cytokines, and the absolute numbers (Ɲ) of CECs and EPCs in children 24 h before and 48 h after CPB surgery to identify high-risk patients of complications. A cohort of 56 children was included: 41 cases with CHD-PAH (22 with high pulmonary flow and 19 with low pulmonary flow) and 15 control cases. We observed that Ɲ-CECs increased in both CHD groups and that Ɲ-EPCs decreased in the immediate post-surgical period, and there was a strong negative correlation between ET-1 and CEC before surgery, along with significant changes in ET-1, IL8, IL6, and CEC levels. Our findings support the understanding of endothelial cell precursors’ role in endogenous repair and contribute to knowledge about endothelial dysfunction in CHD.

## 1. Introduction

Congenital heart disease (CHD) is a structural cardiovascular malformation that is present at birth and continues to represent a significant cause of morbidity and mortality worldwide. Almost 60% of CHD patients experience heart failure within the first year of life, underscoring the importance of early detection and diagnosis in neonates. CHD ranks as the primary cause of death from non-communicable diseases (NCDs) among individuals under 20 years old [1]. The Pediatric Hospital at the National Medical Center Century XXI of the Mexican Social Security Institute in Mexico City provides treatment for approximately 100 children with CHD annually, with the majority of cases involving intraventricular communication and tetralogy of Fallot and requiring cardiac surgery utilizing cardiopulmonary bypass (CPB). CPB triggers an inflammatory response similar to the systemic inflammatory response syndrome, resulting in changes in cardiovascular and pulmonary functions [2,3]. This post-CPB inflammatory response is believed to contribute to postoperative complications and mortality, especially in newborns [4]. The specific cellular and molecular mechanisms involved in this response are not fully understood, but it is thought that the release of cytokines increases the level of endothelin-1 and activates endothelial progenitor cells (EPCs), potentially leading to myocardial diseases and vascular lesions [5]. Pulmonary arterial hypertension (PAH) can be a complication in CHD, and while the exact cause of PAH in CHD is not fully understood [6], endothelial dysfunction is considered to be a significant factor. Among the main modulators in this context are endothelin-1 (ET-1), circulating endothelial cells (CECs), and endothelial progenitor cells (EPCs), which have been suggested as useful biomarkers for the severity and progression of the disease [7,8,9,10,11,12,13].

ET-1 is a peptide derived from vascular endothelial cells (ECs) with vasoconstrictor activity that can lead to the proliferation of vascular smooth muscle cells. It is produced and released into the bloodstream by ECs in the pulmonary vessels [12]. Various stimuli can cause an increase in the synthesis of ET-1 in PAH, making it a significant factor in the development of PAH [14], whereas ET-1 is considered to be an effective vasoconstrictor. Elevated levels of ET-1 are linked with PAH [15]. ET-1 plays a pivotal role in increased vascular tone and vascular remodeling [16]. There is a substantial increase in the expression of ET-1 in the pulmonary vasculature, especially in plexiform lesions in PAH [15]. Moreover, elevated ET-1 plasma levels are closely associated with various indicators of PAH, including right atrial pressure, pulmonary artery oxygen saturation, and pulmonary vascular resistance [17]. The damage to the ECs in PAH worsens the constrictive effects of ET-1, leading to an imbalance in the endothelin system [18] and a reduced ability of the endothelium to release vasodilators [15]. This imbalance and overexpression of ET-1 contribute to increased resistance in the pulmonary blood vessels, partly due to a lack of vasodilators and abnormal remodeling of the pulmonary blood vessels. High levels of ET-1 are also linked to an inflammatory response [19]. The increase in ET-1 plasma levels in PAH may result from increased release of ET-1, reduced clearance of ET-1 by the lungs, or a combination of these mechanisms [15]. However, using ET-1 as a biomarker in PAH patients has several disadvantages. It is considered a low-quality biomarker in PAH due to its limited spread, short half-life (approximately 5 min) [20], and inaccurate representation of the concentration of ET-1 in tissues [15]. Furthermore, ET-1 plasma levels are influenced by various factors, such as ethnicity (e.g., higher ET-1 levels in people of African descent), gender (higher levels in males), and age (higher levels with aging) [21]. Despite the aforementioned points, this study aims to investigate the diagnostic and prognostic value of this and other biomarkers in the blood of children. Structural, functional, and metabolic changes in ECs are believed to be crucial pathological features in pulmonary blood vessels of PAH patients, and ECs dysfunction plays a key role in mediating structural changes in the pulmonary blood vessels and in the development of PAH [22]. Circulating endothelial cells (CECs) are a new marker of endothelial damage found in the peripheral blood under both physiological and pathological conditions [9]. These cells express specific markers and proteins, such as vascular cell adhesion molecule-1, E-selectin, and P-selectin, which may indicate increased coagulation, proliferation, and vasoconstriction. Counting CECs may provide valuable information for monitoring endothelial injury and reflecting the severity of insults to the endothelium. Additionally, circulating cytokines such as IL-6, IL-8, and IL-10 are associated with low survival rates in PAH patients, indicating their potential role in risk-stratifying patients [23]. However, due to their low sensitivity and specificity, they are not routinely used in clinical practice. Furthermore, high levels of circulating cytokines (IL-1β, IL-6, and TNF-α) are involved in the initiation and progression of PAH. Some studies suggest that biomarkers tied to inflammatory and immune processes may play a role in PAH, providing insights into the complex interaction between the immune response and pulmonary vascular remodeling [24]. Increased levels of pro-inflammatory serum cytokines were demonstrated in patients with PAH compared to the control group. Most biomarkers described in the literature are not used in clinical practice, despite being described years ago. For a biomarker to enter clinical use, its utility should be consistently demonstrated in large prospective trials, which are often missing for most identified biomarkers [25]. Limited research has been conducted in this area, making it a priority to measure these markers as a whole in the pediatric population to identify patients at risk of complications and evaluate the prognosis of the disease.

## 2. Results

### 2.1. Study Population

This study involved 56 children, 30 males (53.6%) and 26 females (46.4%), ages ranging from <1 month to 17 years and 11 months. Of the total, 41 underwent scheduled correction surgical procedures via CPB (22 (39.3%) with HPF and 19 (33.9%) with LPF), and 15 (26.8%) corresponded to the control group. Six patients in the CHD group died (three female and three male), of whom four belonged to the CHD HPF group and two were in the CHD LPF group. Three of them died after the first 24 h due to heart failure. The clinical and demographic characteristics of the study population are shown in Table 1.

### 2.2. Statistical Analysis

Correlation strengths between variables were assessed via Spearman rank correlations, as shown in Figure 1.

### 2.3. Plasma Levels of ET-1, CEC, and EPC

A complementary analysis was performed to explain the behavior of endothelial and progenitor cells from the initial mechanism of congenital heart disease and the effect of the surgical procedure, identifying the value of endothelin-1 as the only related variable (Appendix A). A comparative intergroup analysis of consecutive biomarker assessments showed significant differences in plasma ET-1 levels (Figure 2a), with the highest median in the CHD LPF group after surgery, followed by a less high median also in the CHD LPF group after surgery compared to the control group. Differences were also observed in the absolute number of CECs (Figure 2c) and the absolute number of EPCs (Figure 2d) in the CHD LPF group compared to the control group.

When comparing the values of the biomarkers of interest 24 h before and 48 h after the surgical procedure by study group, statistical differences were found (see Table 2).

### 2.4. Flow Cytometry (FC) Analysis

Using this protocol with immune magnetic separation, two populations were detectable via FC. One population consisted of [CD34+/CD146+/VEGFR2+/CD133−] cells (referred to as viable CECs), and the other was represented by [CD34+/CD146+/VEGFR2+/CD133+] cells (referred to as viable EPCs). As shown in Figure 3, once cells were selected according to size and granularity, the cells totally positive for CD146 were selected. In this CD146+ population, the value of median fluorescence intensity (MFI) was determined by the relative expression of CD133. All cells were also positive for VEGF (Appendix A).

### 2.5. Plasma Cytokine Level Differences

As can be seen in Table 2, preoperative levels of all cytokines measured were lower in patients in all three groups. In the intergroup analysis, IL-6 showed the most marked difference in CHD LPF; it only had representative values between CHD LPF and CHD HPF, and the *p* value was 0.001. See Figure 4. 

## 3. Discussion

In our experience, exploratory research encompassing all these markers in the pediatric population is scarce, which suggests that the determination of these biomarkers could help in the clinical evaluation of pediatric patients with CHD HPF or CHD LPF who are candidates for cardiac surgery via CPB. In these patients, contact activation of the blood cells with artificial surfaces, air, surgical trauma, etc. triggers an inflammatory process with cell activation and endothelial dysfunction [26,27]. Regarding the control patient population, the patients selected for the control group were pediatric patients who were operated on for non-cardiac pathologies and underwent surgery for programmed procedures, which guaranteed clinical stability. This represented equality of conditions in the comparison groups in relation to the anesthetic event and in-hospital follow-up and made it possible to clearly establish clinical differentiators such as exposure to extracorporeal shunting and the physiological effect on the pulmonary vasculature generated by the CHD group.

With regard to the population of the CHD with HPF group, it is notable that those patients correspond to the group of heart diseases with the highest prevalence and that the high flow (or precapillary) mechanism generates World Health Organization Group I pulmonary hypertension. Predominantly, these types of pre- and post-tricuspid septal defects generate increased pulmonary vascular resistance and should be previously classified in risk areas to identify those patients who can generate pulmonary hypertension crises and right ventricular dysfunction. In this study, patients with eligibility criteria (pulmonary vascular resistance units less than 3 Wood units) with low risk were selected in order to exclude patients susceptible to complications in the immediate postoperative period who may require advanced maneuvers to improve right ventricular function, such as the use of nitric oxide or drugs. Consequently, all patients were electively extubated without complications, allowing us to represent the best conditions for biomarker measurement without the need to consider potentially confounding variables. In addition, regarding heart disease with reduced pulmonary flow, this group was represented by predominantly truncal cardiac malformations that presented an obstructive mechanism of pulmonary circulation attributed exclusively to obstruction of the pulmonary artery (in any of its portions), and therefore the perfusion of the pulmonary arteries did not depend on aortopulmonary collaterals. Furthermore, the type of surgical repair in all cases involved reconstitution of flow to the pulmonary ventricle with pulmonary physiology, so that tetralogy of Fallot was the most frequent ischemic heart disease. Consequently, these patients exemplified the effects of increased pulmonary flow on biomarkers during the first hours.

In our study, the intragroup analysis showed that the control group had no significant changes after the surgical event, which suggests that extracorporeal shunting and the absence of cardiac intervention did not influence their levels. The CHD group with the greatest changes was the CHD HPF group; this group had previously been diagnosed with pulmonary hypertension, and it was to be expected that during the first 48 h, the amount of flow and pressure to the pulmonary vasculature would decrease. The opposite was the case for the CHD LPF group, which previously maintained an environment of oligohemia and after surgery showed changes generated by the increase in flow and pressure in addition to the fact that the pulmonary arterial tissue was manipulated directly by the surgeon.

Conversely, studies have shown that the bioactive peptide endothelin 1 (ET-1) mediates vasoconstriction of the systemic circulation and influences myocardial contractility [28]. We measured higher baseline values of ET-1 than did Komai et al. [29] and Xia et al. [30], and we assume that this discrepancy cannot be completely explained by variations in the specificity of the monoclonal antibodies used in each study. Although the variations in ET-1 concentrations detected in the plasma of control cases by different researchers may be between 0.1 and 48 fmol/mL (0.25–120 pg/mL) [29], this may reflect patient differences due to age or ethnic origin. The mean age of the patients in Komai’s study was 1.6 years; the patients in Xia’s study were older; and our patients had an intermediary age, so it is possible that ET-1 levels increase in patients with CHD at different ages. In our study, the concentrations of ET-1 in pediatric patients after CPB were much higher than the documented plasma levels, which can be explained by the use of aprotinin for the inactivity of kallikreins in the plasma samples, which limits peptide detection [31,32].

We performed an analysis to explain the behavior of ET-1, and a linear regression analysis was performed to determine whether there was a quantitative explanation for all of the biomarkers measured, plus age and extracorporeal shunt time. Most of these variables were excluded from the first step, and only the Ɲ of CEC and EPC cells had a low degree of correlation. This reflects that there are other variables that were not measured that can be directly attributed to these findings, for example, changes in pulmonary pressure, natriuretic peptide, and right ventricular function.

In this study, CECs increased in both CHD groups and EPCs decreased in the immediate post-surgical period. To explain this cell behavior, like other studies, we also assume that the detection of elevated numbers of CECs may be the most direct marker of endothelial activation or injury and, perhaps, may enable quantification of the inflammatory response in conjunction with CPB. Since these cells are found very rarely in healthy people’s blood, the increased number may reflect the degree of endothelial activation or damage and even represent a prognostic indicator in patients developing an overwhelming inflammatory response after a period of CPB [33,34].

Previous studies have used different protocols for the measurements of CECs and EPCs in pediatric populations with CHD [7,8,9,10,11,12,13], and so far, several assays that allow the detection of CECs and EPCs have been described and are likely to be improved. In addition, we agree with other research groups that it is very important to establish the true value of the enumeration of CECs and EPCs, and that a general consensus on the best way to enumerate these cells is still required [33,34], especially in pediatric populations with CHD, which can be complicated by pulmonary arterial hypertension.

With regard to the results obtained for plasma cytokine levels in this pilot study, inflammatory cytokines are usually altered in pathophysiological states of sepsis and septic shock; in cardiac surgery [35,36], they may increase during the first hours in patients with extracorporeal shunt vascular syndrome or low output syndrome. However, there was no relationship between them in any of the groups, except for IL-6 levels in patients with CHD LPF. This can be explained by the fact that none of the patients underwent surgery with previous hemodynamic instability (all surgeries were elective), they did not develop in-hospital infections during their evolution, and the groups with CHD did not present vascular syndromes secondary to disease. However, CPB with LPF has a higher risk and surgical complexity because it involves ventriculotomy and manipulation of the pulmonary artery, which may explain the increase in IL-6.

Related to the control of biases and disadvantages of the study, and due to the longitudinal nature of the study, misclassification biases were controlled with the selection criteria. Follow-up was limited, and it was not possible to fully determine the changes in biomarkers. There are confounding variables that were not weighted, such as pharmacological management before and after the operation, as well as the influence of other comorbidities. The reduced spectrum of heart diseases included in this study allowed us to establish the influence of pulmonary flow modification but did not allow us to establish other potentially confounding variables. It is possible that the control group could be better represented with patients without disease; however, this was considered unethical. Although this study had a small number of samples and should be considered a limitation, we believe that the results obtained offer the possibility of identifying patients at high risk of complications; however, we do not rule out the need for a study with a larger cohort to make our observations more convincing. To our knowledge, there are no studies of this nature in the pediatric population that include all these biomarkers and their influence on pulmonary hypertension.

## 4. Materials and Methods

### 4.1. Study Population and Inclusion Criteria

The study population comprised children who underwent a scheduled correctional surgical procedure via cardiopulmonary bypass. Patients registered with CHD were divided into two groups: the CHD group with increased pulmonary flow (CHD HPF) and the CHD group with decreased pulmonary flow (CHD LPF). The control group included patients with non-cardiac diseases and non-urgent surgical procedures. The selection criteria for the group with CHD were a) cardiac malformations of an increased pulmonary flow type, i.e., atrial septal defect, ventricular septal defect, anomalous pulmonary vein connection without collector obstruction, and atrioventricular canal without pulmonary stenosis, with a previous echocardiographic study with a moderate to high probability of pulmonary arterial hypertension. The patients with a high probability of PAH had cardiac catheterization with indexed units of pulmonary vascular resistance less than 3 Wood units and a ratio of pulmonary and systemic resistances less than 0; and cardiac malformations of decreased pulmonary flow of the biventricular physiology of the following types: tetralogy of Fallot, double right ventricular outflow tract with pulmonary stenosis, and critical pulmonary stenosis. All patients were extubated during the first 48 h of the procedure. The control group included patients with non-cardiac diseases requiring non-urgent procedures, such as cleft lip correction, ventriculoperitoneal shunt, and urinary tract repair. These patients were extubated upon discharge from the operating room and did not require any type of inotropic amine or vasopressor during the first 72 h. All patients were admitted 24 h before surgery in order to facilitate evaluations by the pediatrics, surgery, and anesthesiology services regarding the clinical conditions of hemodynamic stability and absence of septic foci; other demographic and clinical variables were also recorded. Blood samples: 1.5–4 mL (amount used according to [37] and endorsed by the Ethics Committee of our institution), were collected at two points in time: 24 h pre-surgery and 48 h following surgery completion. Peripheral blood samples were drawn directly into ethylene diamine tetra-acetic acid (EDTA) tubes and divided into two parts to process the biomarker analyses: one part was processed immediately for ET-1 assays, and the other was used for routine analyses in a hematology analyzer CELL-DYN 400 (Abbott Diagnostics, Abbott Park, Chicago, IL, USA) and cell isolation. The remaining sample was centrifuged at 4 °C to obtain plasma aliquoted for preservation at −80 °C for cytokine assays. This comparative cross-sectional study was carried out in a Social Security Pediatric Hospital in Mexico City. Our study was approved by the Research Ethics Committee of the Centro Médico Nacional Siglo XXI (Health Research Coordination), and the approval number is CNIC-R-2010-785-038. Informed consent was requested from the parents of all patients or their primary caregivers, in accordance with the principles outlined in the Declaration of Helsinki (as revised in 2013).

### 4.2. Endothelin-1

The endothelin-1 assay was performed immediately after blood withdrawal. Blood was transferred from the vacutainer tube to chilled, siliconized disposable glass tubes containing aprotinin (1000 kallikrein inactivator units/mL) and gently rocked several times, then centrifuged at 4 °C and the plasma was collected. Subsequently, peptide extraction was performed using HPLC cartridges and a C-18 SEP-COLUMN (Waters Inc., Milford, MA, USA) as the manufacturers recommended, and then a concentrator (Vacufuge™, Eppendorf, Hamburg, Germany) was used to dry samples, which were subsequently freeze-dried overnight using a lyophilizer and stored at −70 °C until the radioimmunoassay (RIA) was performed. Plasma ET-I concentration was measured with an RIA kit (Phoenix Pharmaceuticals, Inc., Burlingame, CA, USA) according to the manufacturer’s protocols. Briefly, the samples were reconstituted in an RIA buffer. ET-1 labeled with I125 was used as tracer, and porcine ET-1 was used as a standard. The assay was incubated at 4 °C in 0.1 mol/L phosphate buffer, at pH 7.4, containing 0.1% BSA and 0.1% Triton-X-100. Bound and free fractions were separated using a secondary antibody. Recovery from the extraction procedure was 85% ± 5% on the basis of spiked plasma standards (4–20 fmol/mL). Samples were analyzed in duplicate, and the results were averaged.

### 4.3. Circulating Endothelial Cell and Endothelial Progenitor Cell Isolation and Counts

Peripheral blood mononuclear cells from blood taken from patients were isolated with the use of a density gradient with a Ficoll-paque PLUS (GE17-1440-02, Amersham Biosciences, Piscataway, NJ, USA), according to standard protocols. Recovered cells were washed (10 mL of cold PBS containing 0.5% (*w*/*v*) bovine serum albumin (BSA) and 1.5 mM EDTA), and the pellets were resuspended in 500 µL of PBS and filtered through a 40 µm cell strainer. The viability of the cells was determined using a Trypan blue dye (Sigma-Aldrichs, St. Louis, MO, USA) exclusion test. Initially, isolated cells were counted manually in a Neubauer chamber (Marienfeld, Germany) under an optical microscope and suspended at a concentration of 2 × 10^7^ cells/mL. Afterwards, absolute cell counts were obtained by flow cytometry using commercial fluorescent-counting beads from BD Biosciences, Pharmigen (San Jose, CA, USA), added to the stain tube for immunophenotyping. At least 100,000 events were captured.

#### 4.3.1. Antibodies and Reagents

Biotinylated anti-human MCAM/CD146 antibody was obtained from R&D Systems, Inc. (Minneapolis, MN, US); Ig type: goat IgG (AB_10050580, BD™ IMag, San Jose, CA, USA) and phycoerythrin (PE)-labeled anti-CD146 monoclonal antibody, clone P1H12, mouse IgG1, κ (2 µL/10^5^ cells; 1:100 dilution); allophycocianin (APC)-labeled anti-CD34 monoclonal antibody, clone 8G1, mouse IgG1, κ (20 µL/10^5^ cells; 1:10 dilution); fluorescently labeled isotype-matched IgG1 antibodies, and BD Mouse Fc Block™ CD16/CD32 mAb 2.4G2 were purchased from BD Biosciences, Pharmigen (San Jose, CA, USA). Cyanine dye Cy5-labeled anti-CD133 was obtained from Miltenyi Biotec (25 µg/10^6^ cells, 1:100 dilution); fluorescein isothiocyanate (FITC)-labeled anti-VEGFR2 monoclonal antibody, clone 89106; and mouse IgG1 were purchased from R&D Systems, Inc. (10 µL/105 cells, 1:10 dilution). This combination of fluorochromes allowed the staining of each sample to measure the expression of antigens on CD146+ cells. PE-labeled mouse IgG1 and FITC-labeled mouse IgG2a were used as isotype-matched controls. In order to minimize unspecific staining using antibody binding via Fc-receptors, all samples were treated with a commercially available BD Mouse Fc Block™, BD Biosciences, Pharmigen (San Jose, CA, USA). Red blood cells were lysed using a commercially available reagent (lysing solution IO-Test3, Beckman Coulter, Pasadena, CA, USA) For all staining and washing steps, cells were kept in PBS supplemented with BSA (0.5%) and EDTA (2.0 mM).

#### 4.3.2. Immunomagnetic Separation and Cell Counting

One aliquot of the absolute count was resuspended in cell staining buffer at a concentration of 2 × 10^7^ cells/mL, plus BD Mouse Fc Block™, and incubated on ice for 15 min; this was used for the immunocapture of endothelial cells. Immunomagnetic separation was performed at 4 °C with magnetic beads (BD™ IMag) coated with S-Endo 1 biotinylated anti-human MCAM/CD146 antibody, a monoclonal antibody created against the endothelial antigen CD146. To avoid nonspecific binding of leukocytes to the CD146-coated microspheres, cell suspensions were rinsed vigorously through the pipette tip during the washing steps. Cells were then isolated with a BDI magnet via positive selection with streptavidin-Plus particles according to the manufacturer’s recommendations and measured via FC. Briefly, the cells were incubated for 30 min at room temperature with fluorochrome-conjugated antibodies αCD34/APC, αCD133/APC, αVEGF/FITC, and αCD146/PE at the concentrations recommended by the manufacturer. Fixing and lysing solution (FLS, Becton Dickinson, San Jose, CA, USA) was added (100 µL), and samples were incubated for 10 min. PBS was added (2 mL), and tubes were centrifuged at 1200× *g*/5 min. Cell pellets were resuspended in PBS and analyzed using a FACS Aria Flow Cytometer with DIVA Software v 6.1. A total of 100,000 total events were captured. The algorithm for endothelial identification was as follows: cells were first selected based on a forward vs. side scatter pattern. From CD146+ cells, gates were made for VEGF+/133+ or VEGF+/133−, and VEGF+/CD34+ or CD34−. Percentage values were obtained, and cell quantification was calculated based on total PMCs and total leucocyte numbers. The absolute CEC and EPC numbers (N) were derived from the absolute numbers of white blood cells provided by a hematology analyzer and the percentage of CECs and EPCs was determined via FC using the following formula [38]: percentage of cells × white blood cell (WBC) count/100.

### 4.4. Cytokine Quantification

The concentrations of TNF-alpha, IL-1B, IL-6, IL8, IL-10, and IL-12p70 were determined using the Human Inflammatory Cytokine kit-Cytometric Bead Array (BD Biosciences, San Jose, CA, USA), which is a system for multiple cytokine quantification. According to the manufacturer’s recommendations, 100 µL of plasma samples or standard controls for the standard curve were incubated at room temperature with recovered fluorescent beads with antibody capture with specificity corresponding to the fluorescence intensity. After 3 h, wash buffer (200 µL) was added, and samples were centrifuged for 5 min at 900× *g*. After the supernatant was discharged, phycoerythrin (PE)-conjugated cytokine detection antibodies were added. Thirty minutes later, the beads were washed twice and resuspended in phosphate-buffered saline (PBS) (100 µL) and analyzed using an FACS-Aria flow cytometer with Diva v 6.1 software (BD, Becton Dickinson, San Jose, CA, USA). Using the software provided by the manufacturer, from the FMI values for PE, the cytokine concentration (in pg/mL) was determined.

### 4.5. Statistical Analysis

Clinical and demographic data were collected and analyzed using descriptive statistics according to the distribution of the data. Absolute frequencies and percentages were used for categorical variables (among types of CHD); the continuous variables did not present a parametric distribution and were therefore expressed as medians and interquartile ranges (25th and 75th percentiles). Inferential statistics were performed in the following stages: (a) comparison of biomarkers between the three groups (control, CHD HPF, CHD LPF) with nonparametric related samples with a Wilcoxon rank test; (b) secondary analysis was performed to explain the behavior of endothelial cells with a linear regression test. All tests were two-sided, with alpha set at *p*-value ≦ 0.05. Patients with missing data were excluded from the analysis. Data analyses were performed using SPSS version 22 statistical software by IBM (Chicago, IL, USA).

## 5. Conclusions

The results revealed that ET-1, CEC, EPC, and IL-6 are significantly correlated in children with CHD-PAH. To validate these data, more studies should be conducted and the sample size increased in this population. ET-1, CEC, EPC, and IL-6 had the highest values, suggesting their diagnostic and prognostic use in pediatric patients in the immediate postsurgical period.

## Figures and Tables

**Figure 1 ijms-25-08895-f001:**
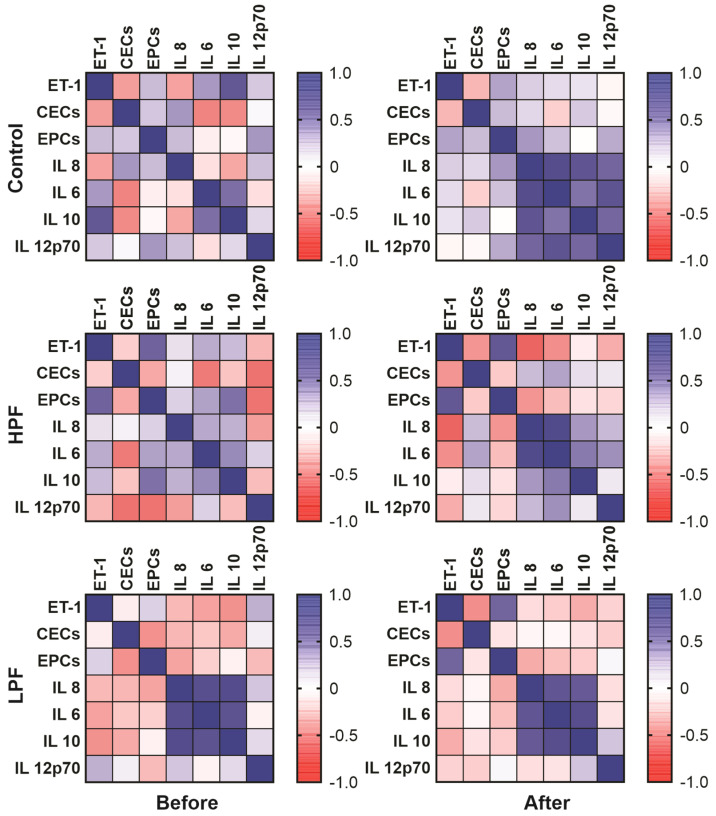
Heatmap of corresponding Spearman correlation coefficients for the variables analyzed before and after surgery, where the positive correlation (directly proportional) is shown in blue and the negative correlation (inversely proportional) is shown in red, according to the color scale on the side of each subpanel. HPF corresponds to the group with high pulmonary flow, and LPF corresponds to the low pulmonary flow group. *p* values are presented in Table 2.

**Figure 2 ijms-25-08895-f002:**
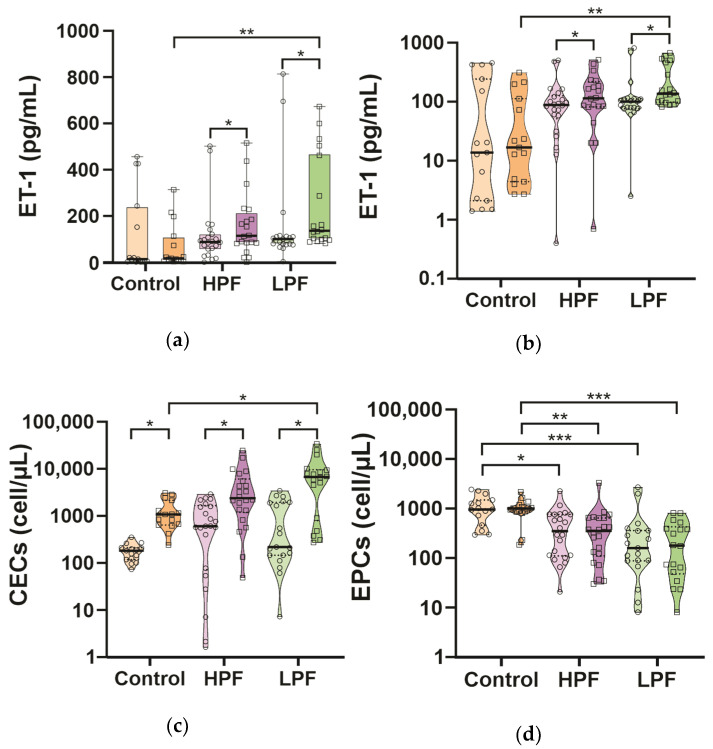
Comparative intergroup analysis of ET-1 plasma levels, CECs, and EPCs. Subpanel (**a**) shows the comparison of plasma ET-1 levels in the three groups of patients before and after surgery. Subpanel (**b**) shows the intergroup comparison as a violin graph with a logarithmic scale for ET-1. Subpanels (**c**,**d**) correspond to intergroup comparisons of the absolute numbers of CECs and EPCs, respectively, with a logarithmic scale, before and after surgery. Statistically significant: * *p* ≤ 0.05; ** *p* ≤ 0.01; *** *p* ≤ 0.001 (post hoc Bonferroni test). The fainter color and empty circles in each group represent measurements 24 h before surgery, while the more intense color and empty squares in each group indicated measurement 48 h after surgery. In all subpanels the medians in each group are represented by a thicker black line.

**Figure 3 ijms-25-08895-f003:**
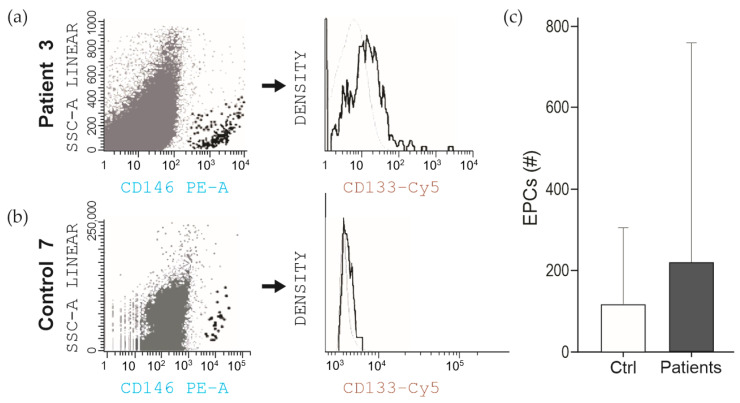
Flow cytometric detection of endothelial cells. Identification of EPCs by expression of CD46 and CD133 in representative plots from patient (n = 3), panel (**a**), and control (n = 7) panel (**b**). Most cells were also positive for VEGF (Appendix A). Panel (**c**): Quantitative analysis showed a greater proportion of endothelial cells was observed in CHD patients than in the control group.

**Figure 4 ijms-25-08895-f004:**
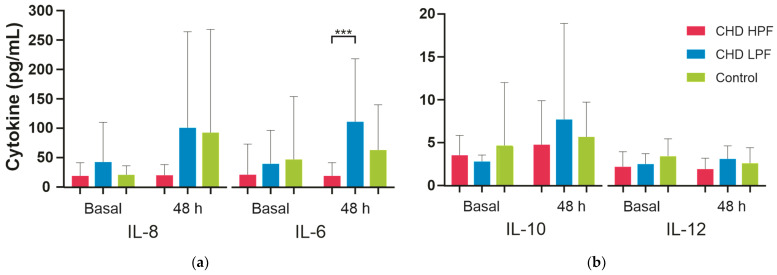
Plasma cytokine levels measured via flow cytometry in patients and controls. Panel (**a**) shows the comparison of plasma levels of IL-8 and IL-6 cytokines measured using flow cytometry in the three groups of patients, where it can be seen that the highest values of both cytokines were obtained in CHD LPF group. Panel (**b**) shows the comparison of plasma levels of IL-10 and IL12p70 in the three groups of patients before and after surgery; it can be seen that the highest levels of IL-10 are in the control group, followed by the CHD LPF group after surgery. Statistically significant: *** *p* ≤ 0.001.

**Table 1 ijms-25-08895-t001:** Demographics and clinical characteristics of all study groups.

	CHD HPF	CHD LPF	Control
n (%),	22 (39.3)	19 (33.9)	15 (26.8)
Sex: M, male/F, female	10 M/12 F	11 M/8 F	9 M/6 F
CHD Type			
Ventricular septal defect	16 (72.7)	-	-
Atrial septal defect	3 (13.6)	-	-
Anomaly connection pulmonary veins	2 (9.1)	-	-
Atrioventricular defect	1 (4.6)	-	-
Fallot Tetralogy	-	13 (68.4)	-
Pulmonary stenosis	-	4 (21.0)	-
Ebstein anomaly	-	2 (10.2)	-
Median, (p25–p75)			
Age (months): 32 (21–60)			
Heigh (cm): 90 (74–107)			
Weight (kg): 12.5 (8.2–17.2)			
Surgical time (min): 92.4 (72–128.4)			
Aortic impingement (min): 64.2 (33.6–75.6)			

HPF, high pulmonary flow; LPF, low pulmonary flow; p25, percentile 25; p75, percentile 75.

**Table 2 ijms-25-08895-t002:** Analytical data for CHD patients and controls.

	Control		CHD HPF		CHD LPF		Wilcoxon Test (*p*-Value)			Kruskal Wallis (*p*-Value)	
	Before (a)	After (b)	Before (c)	After (d)	Before (e)	After (f)	a vs. b	c vs. d	e vs. f	Before	After
ET-1 (pg/mL)	13.8 (1.4–456.6)	16.8 (2.7–313.4)	90.5 (0.4–813.9)	131.8 (0.7–672.5)	87.6 (0.4–813.9)	131.8 (0.7–672.5)	0.16	0.01	0.04	0.114	0.002
IL-8 (pg/mL)	17.4 (5.4–49.2)	18.8 (2.6–647.1)	9.2 (2.6–187.9)	15.9 (2.6–673.5)	9.1 (2.6–187.9)	21.5 (4.9–673.5)	0.334	0.560	0.267	0.529	0.101
IL-1B (pg/mL)	6.2 (6.2–6.2)	6.2 (6.2–10.3)	6.2 (6.2–8.6)	6.2 (6.2–6.2)	6.2 (6.2–8.6)	6.2 (6.2–6.2)	0.317	1.000	0.317	0.398	0.282
IL-6 (pg/mL)	12.8 (1.5–424.1)	28.7 (1.5–209.3)	5.2 (1.5–232.7)	31.5 (1.5–401)	5.8 (1.5–141.2)	80.4 (1.5–401)	0.209	0.397	0.023	0.278	0.001
IL-10 (pg/mL)	2.3 (2.3–30.9)	3.8 (2.3–14.8)	2.3 (2.3–10.5)	3.7 (2.3–41.3)	2.3 (2.3–8.3)	3.8 (2.3–41.3)	0.139	0.477	0.022	0.866	0.418
TNF-alpha (pg/mL)	2.7 (2.7–2.7)	2.7 (2.7–2.7)	2.7 (2.7–3.8)	2.7 (2.7–2.7)	2.7 (2.7–2.7)	2.7 (2.7–2.7)	1.000	0.317	1.000	0.427	1.00
IL12p70 (pg/mL)	3 (0.9–7.5)	2.3 (0.9–6)	2.3 (0.9–6.4)	2.3 (0.9–6)	2.3 (0.9–5.2)	2.4 (0.9–6)	0.158	0.689	0.167	0.143	0.093
Hemoglobin (g/dL)	12.2 (6.9–15.9)	13.4 (9.2–15.2)	14.4 (10.1–24)	13.8 (9–17.6)	15.0 (10.1–24)	13.6 (9–17.6)	0.243	0.067	0.017	0.007	0.751
Hematocrit (%)	36 (22–47)	39 (26–47)	42 (24–72)	39 (25–78)	44 (24–72)	39.5 (25–78)	0.314	0.015	0.067	0.017	0.687
RBC (cells 10^6^/mL)	3.6 (2.9–5.6)	4.3 (3.2–5.4)	5 (4–10.1)	4.7 (3–6.3)	5 (4.1–10.1)	4.6 (3–6.3)	0.057	0.068	0.003	<0.001	0.449
Platelets (cells 10^3^/mL)	210 (45–493)	247 (95–456)	269 (50–478)	167 (60–620)	233 (127–350)	165.50 (60–620)	0.691	<0.001	0.163	0.004	0.216
Lymphocytes (cells 10^3^/mL)	3.3 (1.2–6.4)	3.4 (1.5–8.5)	2.7 (1.4–5.9)	2.55 (1.1–9.9)	2.6 (1.8–5.9)	2.1 (1.1–9.9)	0.798	0.575	0.349	0.240	0.101
Neutrophil (cells 10^3^/mL)	7 (2–13.7)	4.9 (2–12.2)	5 (2.6–12.7)	6.8 (1.8–16.5)	4.6 (3.1–10.2)	5.4 (2.3–16.5)	0.041	0.247	0.150	0.231	0.221
WBC (cells 10^3^/mL)	1.1 (4.5–20.8)	8.7 (6–25.5)	8.5 (5–17.8)	10.25 (4.1–27.2)	8.4 (5.1–17)	7.8 (4.9–27.2)	0.875	0.372	0.828	0.240	0.678
MNC (cells 10^3^/mL)	3.9 (2–7)	4.2 (2.9–9.3)	3.3 (1.8–6.7)	3.2 (1.3–10.7)	3.2 (2–6.7)	2.8 (1.3–10.7)	0.182	0.903	0.276	0.297	0.005
CEC (cells/μL)	181.8 (73.5–352.9)	1069.5 (240.1–3078.8)	568.18 (1.6–3443.8)	4341.4 (48.8–33,860.6)	421.9 (7.3–3443.8)	6724.5 (273–33,860.6)	<0.001	<0.001	<0.001	0.068	0.010
EPC (cell/μL)	959.4 (295.3–2439.5)	1004.6 (186.4–2167.7)	244 (8.1–2694.1)	294 (8.1–3296.6)	160 (8.1–2694.1)	220.8 (8.1–815.7)	0.69	0.86	0.74	<0.001	<0.001

CHD, congenital heart disease; HPF, high pulmonary flow; LPF, low pulmonary flow; before, 24 h pre-surgery; after, 48 h post-surgery; ET-1, endothelin-1; IL-8, interleukin-8; IL-1B, interleukin 1B; IL-10, interleukin 10; IL-6, interleukin-6; TNF-alpha, tumor necrosis factor-alpha; IL-12p70, interleukin 12p70; RBCs, red blood cells; WBCs, white blood cells; MNCs, mononuclear cells; CECs, circulating endothelial cells identified as [CD34+/146+/VEGF+/133−]; EPCs, endothelial progenitor cells identified as [CD34+/146+/VEGF+/133+].

## Data Availability

Data are contained within the article and Appendix A.

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
