# Peer review of "Levels of Plasma Endothelin-1, Circulating Endothelial Cells, Endothelial Progenitor Cells, and Cytokines after Cardiopulmonary Bypass in Children with Congenital Heart Disease: Role of Endothelin-1 Regulation"

_ijms, 2024, doi:10.3390/ijms25168895_

Round 1
Reviewer 1 Report
Comments and Suggestions for Authors
1. In Figure 2, graph legend is missing.
2. In figure 3, it will be good to show the percentage of CD146 population in a bar graph. Also it looks like the CD146 cell count is very low to consider as good experimental technique.
3. In table 1: CHD Type could be removed
4. The heading of table 2 is on page 5, it should be moved to page 6. So that the table will be complete.
5. Figure 4 is not mentioned in any part of the results section. If it is not described in the results, then it should be moved to supplements.
6. Dilutions of antibodies used for Flow cytometry were not mentioned.
Author Response
|
Response to Reviewer 1 Comments
|
||
|
1. Summary |
|
|
|
Thank you very much for taking the time to review this manuscript. Please find the detailed responses below and the corresponding revisions/corrections highlighted/in track changes in the re-submitted files. |
||
|
2. Questions for General Evaluation |
Reviewer’s Evaluation |
Response and Revisions |
|
Does the introduction provide sufficient background and include all relevant references? |
Can be improved |
Thank you for your comments. We have the introduction re-written and also included some references relevant to our topic of interest. The changes are highlighted in color in the Word file under “Track Changes”.
|
|
Is the research design appropriate? |
Can be improved |
Thank you for such an accurate comment. However, as we mentioned in the limitations of this study, this is a pilot study, and we think that for future research the design will necessarily have to be improved.
|
|
Are the methods adequately described? |
Yes |
Thank you for your comments |
|
Are the results clearly presented? |
Yes |
Thank you for your comments |
|
Are the conclusions supported by the results? |
Can be improved |
Thank you for your comments. We have improved the conclusions to better support the results obtained. The changes are highlighted in color in the Word file under “Track Changes”.
|
|
3. Point-by-point response to Comments and Suggestions for Authors.
|
||
|
Comments 1: In Figure 2, graph legend is missing. |
||
|
Response 1: Thank you for pointing this out. Therefore, we have included the legend to Figure 2 and also included the statistical significance in the figure. this change can be found – page number 6, figure 2 legend and line 167.
|
||
|
Comments 2: In figure 3, it will be good to show the percentage of CD146 population in a bar graph. Also, it looks like the CD146 cell count is very low to consider as good experimental technique. |
||
|
Response 2: We agree with this comment and for more clarity, figure 3 has being modify. With dot plots showing CD146 and CD133 and a panel C was added with the quantitative analysis the total amount of endothelial cells is shown. this change can be found – page number 8, figure 3 legend, and line 193-196. Additionally in supplementary material 2, which was modified.
|
||
|
Comments 3: In table 1: CHD Type could be removed |
||
|
Response 3: We agree with this comment and in Table 1 we have removed: CHD Type.
|
||
|
Comments 4: The heading of table 2 is on page 5, it should be moved to page 6. So that the table will be complete. |
||
|
Response 4: Thanks for the comment, we have moved the heading of table 2 to the next page which now correspond to page 7 so that the table is complete. this change can be found – page number 7, table 2 heading, and line 174.]
|
||
|
Comments 5: Figure 4 is not mentioned in any part of the results section. If it is not described in the results, then it should be moved to supplements. |
||
|
Response 5: Thank you for pointing this out. We agree with this comment. Therefore, we have now described Figure 4 in the results section. and according to another reviewer's comment we have also improved Figure 4. this change can be found – page number 8, paragraph 2.5, and line 197.]
|
||
|
Comments 6: Dilutions of antibodies used for Flow cytometry were not mentioned. |
||
|
Response 6: Agree and we now added both the concentration and dilution used for each of the antibodies used for flow cytometry at the materials and methods 4.3.1. section. this change can be found – page number 12, paragraph 3, and lines 390-397.]
|
||
|
5. Additional clarifications |
||
|
The document was corrected to avoid plagiarism. |
||
Reviewer 2 Report
Comments and Suggestions for Authors
The authors found that CECs increased in both congenital heart disease groups and EPCs decreased in the immediate post-surgical period. The findings are impressive and overall they correspond to contemporary opinion on the role of endothelial cell precursors in endogeneous reparation. I am passionate about the study and congratulate the authors on it. However, I would like to propose the only concern:
1. Overlap between number of CECs and EPCs seems to be interpreted as a reduction in the pool of precursors with reparative potential. In this vein, the authors could consider the prospects for reversal of pulmonary hypertension in the remote period.
Author Response
|
Response to Reviewer 2 Comments
|
||
|
1. Summary |
|
|
|
Thank you very much for taking the time to review this manuscript. Please find the detailed responses below and the corresponding revisions/corrections highlighted/in track changes in the re-submitted files. |
||
|
2. Questions for General Evaluation |
Reviewer’s Evaluation |
Response and Revisions |
|
Does the introduction provide sufficient background and include all relevant references? |
Yes |
Thank you for your comments. |
|
Are all the cited references relevant to the research? |
Yes |
Thank you for your comments. |
|
Is the research design appropriate? |
Yes |
Thank you for your comments. |
|
Are the conclusions supported by the results? |
Yes |
Thank you for your comments. |
|
3. Point-by-point response to Comments and Suggestions for Authors.
|
||
|
Comments 1: Overlap between number of CECs and EPCs seems to be interpreted as a reduction in the pool of precursors with reparative potential. In this vein, the authors could consider the prospects for reversal of pulmonary hypertension in the remote period. |
||
|
Response 1: Thank you for pointing this out. Thank you for this important comment. To which we comment that the objective of the study through the endothelial cell count was diagnostic, to identify patients at high risk of complications and death, and on this research, it was not done for therapeutic purposes, as we commented in our limitations, we will have to perform other studies aimed at treatment, specifically in patients with pulmonary hypertension.
|
||
|
|
||
|
5. Additional clarifications |
||
|
To avoid plagiarism, the manuscript was reviewed and corrected. |
||
Reviewer 3 Report
Comments and Suggestions for Authors
The study by Rangel-Lopez investigated the parameters of endothelial dysfunction and the levels of cytokines in the children with congenital heart disease. The parameters have been compared between control and disease groups with high pulmonary flow (HPF) and with low pulmonary flow (LPF). The authors have found that the number of circulating endothelial cells and ET-1 were higher in the LPF group than in the HPF and control groups. Moreover, the differences in ET-1, IL8, IL6, and CEC levels were significantly changed.
The study has several limitations which were mentioned by the authors. The manuscript should be also improved. Nevertheless, the results provided by the authors could contribute to our knowledge about endothelial dysfunction by CHD.
Comments
1. The abstract should be improved.
2. I also suggest to modify the title: “preliminary” could be avoided, for example “The regulation of Endothelin-1, etc. by…..”
3. What is the main hypothesis of the study?
4. What are the main conclusions of the study? What is new?
5. Figure 4 is not mentioned in the text. Were the differences between the groups statistically significant? Please indicate.
6. In the methods, the amount of blood samples in ml should be added.
Author Response
|
Response to Reviewer 3 Comments
|
||
|
1. Summary |
|
|
|
Thank you very much for taking the time to review this manuscript. Please find the detailed responses below and the corresponding revisions/corrections highlighted/in track changes in the re-submitted files. |
||
|
2. Questions for General Evaluation |
Reviewer’s Evaluation |
Response and Revisions |
|
Does the introduction provide sufficient background and include all relevant references? |
Can be improved |
Thank you for your comments. We have the introduction re-written and also included some references relevant to our topic of interest. The changes are highlighted in color in the Word file under “Track Changes”. |
|
Is the research design appropriate? |
Yes |
Thank you for your comment. |
|
Are the methods adequately described? |
Can be improved |
Thank you for your comments. We have improved the methods section in accordance with the accurate observations you made. |
|
Are the results clearly presented? |
Must be improved |
Thank you for your comments. We have improved the results section in accordance with the observations you made. |
|
Are the conclusions supported by the results? |
Can be improved |
Thank you for your comments. We have improved the conclusions to better support the results obtained. The changes are highlighted in color in the Word file under “Track Changes”.
|
|
3. Point-by-point response to Comments and Suggestions for Authors.
|
||
|
Comments 1: The abstract should be improved. |
||
|
Response 1: Thank you for pointing this out. Therefore, we have re-written and improved the abstract for more clarity. The changes are highlighted in color in the Word file under “Track Changes”.
|
||
|
Comments 2: I also suggest to modify the title: “preliminary” could be avoided, for example “The regulation of Endothelin-1, etc. by…..” |
||
|
Response 2: We agree with this comment and the title has been modified according to your suggestion.
|
||
|
Comments 3: What is the main hypothesis of the study? |
||
|
Response 3: On the basis that cardiopulmonary bypass surgery, currently available for corrective surgery, induces endothelial dysfunction, regulated by endothelin-1, in which circulating endothelial cells and endothelial progenitor cells play a crucial role. Our hypothesis is to assess that these markers can be used to gauge disease severity and progression, which could have immediate postoperative prognostic value. The paragraph was already included in the manuscript.
|
||
|
Comments 4: What are the main conclusions of the study? What is new?. |
||
|
Response 4: Thank you for the comment, although our sample was small, The data obtained correspond to the contemporary view on the role of endothelial cell precursors in endogenous repair and could contribute to our knowledge on endothelial dysfunction due to CHD, additionally the results of these markers analyzed together suggest their possible prognostic use, however it is necessary to increase the sample size to obtain more conclusive results. The paragraph was already included in the manuscript.
|
||
|
Comments 5: Figure 4 is not mentioned in the text. Were the differences between the groups statistically significant? Please indicate. |
||
|
Response 5. Thank you for pointing this out. We agree with this comment. Therefore, we have now have included in the figure the statistically significant difference found between the study groups.
|
||
|
Comments 6: In the methods, the amount of blood samples in ml should be added. |
||
|
Response 6: In agreement and now we add the amount of blood sample in ml and in the materials and methods section 4.1. we have also incorporated a bibliographic reference to further support this point, this change can be found – page number 11, and line 341., reference number 37.
|
||
|
5. Additional clarifications |
||
|
The manuswcritp was reviewed and corrected for plagiarism. |
||